# Equivariant Quantum Neural Networks for Image Clasification

## Abstract

We propose an Equivariant Quantum Neural Network (EQNN) architecture that leverages symmetries commonly present in image data, specifically roto-reflection symmetries. By incorporating symmetries such as rotations and reflections into the quantum neural network's design, we can significantly reduce the number of trainable parameters, thereby decreasing the model's complexity and improving its efficiency. This method enhances learning capabilities with smaller datasets while also promoting better generalization. We evaluate the performance of our model using standard benchmark datasets for image classification and compare it against other quantum models.

## 1   Introduction

In the fast-evolving field of machine learning, incorporating symmetries into model architectures has proven to be a highly effective method for introducing inductive biases. These biases play a key role in improving both how efficiently models are trained and how well they generalize to new data. Symmetry integration allows models to better utilize the intrinsic patterns within the data, thereby reducing the need for large datasets and extensive pre-processing. Geometric machine learning has shown that the incorporation of symmetries in models significantly simplifies optimization tasks, leading to faster training and improved performance in a wide range of applications.

In recent years, the combination of quantum computing and geometric machine learning has given rise to a new subfield called geometric quantum machine learning (GQML), which brings symmetries into quantum model architectures. A promising development in this area is the use of Equivariant Quantum Neural Networks (EQNNs), which have shown potential in overcoming challenges unique to quantum computing. One such challenge is the barren plateau problem, which hinders optimization in quantum circuits. EQNNs aim to preserve symmetry while leveraging the power of quantum computing, opening new possibilities for tackling tasks such as image classification and pattern recognition.

This work focuses on embedding roto-reflection symmetries into quantum convolutional neural networks (QCNNs) to create equivariant quantum convolutional neural networks (EQCNNs). These models are specifically designed to be equivariant under geometric transformations, such as 90° rotations and reflections over both the X and Y axes. By embedding these symmetries directly into the architecture, we aim to improve the model's capacity for recognizing patterns and classifying images more accurately and efficiently while reducing the need for large training datasets compared to models that do not leverage symmetry.

Equivariant models offer several key advantages. First, by reducing the number of parameters needed in the model, they streamline the learning process, making training faster and less computationally demanding. This reduction in parameters also helps prevent overfitting, ensuring that

Submitted to the Second Workshop on Machine Learning with New Compute Paradigms at NeurIPS (MLNCP 2024). Do not distribute.

the model does not memorize specific details of the training data but instead generalizes well to unseen data. Additionally, because the model is designed to be invariant to certain symmetries in the data, it reduces the number of possible outputs, allowing the model to learn more efficiently even with limited data, and without requiring data augmentation techniques. Another benefit is weight-sharing, which further reduces the number of parameters that need to be optimized, leading to improved computational efficiency.

Despite these advantages, there are important considerations when using equivariant models. A critical challenge is ensuring that the data itself reflects the symmetries incorporated into the model. If the data does not exhibit these symmetries, the model's expressivity will be limited, potentially leading to suboptimal training outcomes. In such scenarios, enforcing equivariance may constrain the model's ability to learn effectively, as it would be restricted to a space that does not align with the true structure of the data. Therefore, it is crucial to ensure alignment between the symmetries embedded in the model and the characteristics of the data being used.

## 1.1 Quantum Machine Learning

Quantum Machine Learning (QML) is an emerging field at the intersection of quantum computing and machine learning. QML seeks to utilize the unique properties of quantum systems, such as superposition and entanglement, to potentially surpass classical machine learning algorithms in terms of speed and efficiency, particularly on noisy intermediate-scale quantum devices (NISQ).

One of the most widely studied approaches in QML is the Quantum Neural Network (QNN), which is a quantum counterpart to classical neural networks. QNNs are typically implemented using Variational Quantum Algorithms (VQAs), which combine quantum circuits with trainable parameters optimized through classical feedback loops.

The main components of a QNN include the following:

- Data Embedding: A classical input is mapped into a quantum state through a quantum feature map $\phi : X \to H$, where $H$ is a Hilbert space and $x \to |\phi(x)\rangle$ represents a classical input transformed into a quantum state via a unitary operation $U_\phi(x)$.

- Ansatz (Variational Quantum Circuit): A variational quantum circuit consists of quantum gates with trainable parameters that are adjusted during the training process to optimize the model. Typically, these circuits use rotation gates that apply tunable rotations to qubits.

- Measurement: Once the quantum state is prepared, one or more qubits are measured to obtain the output. The measurement is typically performed with respect to the Pauli-Z observable, yielding expectation values that contribute to the final prediction.

In a QNN, predictions are obtained by measuring the expectation values of certain observables:

$$y(\mathbf{x}) = \langle \psi(\theta, x)|O|\psi(\theta, x)\rangle \tag{1}$$

These hybrid quantum-classical models have demonstrated promising results in various applications, offering a potential solution to quantum machine learning's scalability and trainability issues.

## 1.2 Equivariant Quantum Neural Networks

An Equivariant Quantum Neural Network (EQNN) is a type of QNN designed to respect the symmetries present in the data. For image classification tasks, incorporating roto-reflection symmetries (such as 90° rotations and reflections) can reduce the model's complexity by ensuring that the output remains invariant under these transformations.

To build an EQNN, each component of the QNN (data embedding, ansatz, and measurement) must satisfy the symmetry conditions. Specifically, an equivariant embedding transforms classical data into quantum states that reflect the symmetry of the dataset. The ansatz is designed using quantum gates that respect these symmetries, and the measurement is carried out with respect to an invariant observable. The objective is to ensure that the model's output remains unchanged under symmetry transformations, i.e.,

$$\begin{aligned} y_\theta(g[x]) &= \langle\psi(g[x])|\mathcal{U}^\dagger(\theta)O\mathcal{U}(\theta)|\psi(g[x])\rangle \\ &= \langle\psi(x)|V_g^\dagger\mathcal{U}^\dagger(\theta)O\mathcal{U}(\theta)V_g|\psi(x)\rangle \\ &= \langle\psi(x)|\mathcal{U}^\dagger(\theta)(V_g^\dagger OV_g)\mathcal{U}(\theta)|\psi(x)\rangle \\ &= \langle\psi(x)|\mathcal{U}^\dagger(\theta)O\mathcal{U}(\theta)|\psi(x)\rangle \\ &= \langle\psi(\theta,x)|O|\psi(\theta,x)\rangle = y_\theta(x), \, \forall x \in \chi, \, \forall g \in G. \end{aligned}$$

## 2  Method

### 2.1  Data

**MNIST**: We utilize the MNIST dataset, which is a widely used benchmark in the field of image classification. It contains 70,000 images of handwritten digits (0–9) along with their corresponding labels. In this study, we focus only on two classes, specifically the digits 0 and 1.

**Fashion-MNIST**: Fashion-MNIST is another widely adopted dataset, consisting of grayscale images of Zalando's articles of clothing. It contains 60,000 training examples and 10,000 test examples, each labeled from one of 10 clothing categories. For this work, we preprocess the images to 16x16 pixels and restrict our focus to just two classes: T-Shirts (class 0) and Trousers (class 1).

For both data, we use (16,16,1) normalized images.

### 2.2  Roto-Reflection Equivariant Quantum Neural Network

Our proposed Equivariant Quantum Convolutional Neural Network (EQCNN) incorporates symmetries such as 90° rotations and reflections along the X and Y axes. These symmetries are frequently encountered in image datasets, and our goal is to design a model architecture that respects these transformations. The key components of our EQCNN are described below:

#### 2.2.1  Equivariant Quantum Embedding

We utilize the Coordinate-Aware Amplitude (CAA) embedding [1], which explicitly encodes the x and y coordinates of each pixel. The x-coordinate is represented by the first set of qubits, and the y-coordinate by the second set. The embedding of an image $x_{ij}$ into a quantum state is given by:

$$|\psi(x)\rangle = \sum_{i=0}^{N-1}\sum_{j=0}^{N-1} x_{ij}|i\rangle|j\rangle \tag{2}$$

where $N$ is the size of the image. This quantum embedding maps an image into a vector of $N^2$ elements, which is subsequently encoded into a quantum circuit using amplitude embedding.

The key idea with this embedding is to satisfy the equivariant data embedding condition

$$|\psi(g[x])\rangle = V_g|\psi(x)\rangle \tag{3}$$

Where $g \in G$ is the symmetry operation $g$ and $V_g$ is a unitary operator corresponding to this symmetry that acts over a quantum state.

In this sense, we can find the induced representation of the symmetries such as reflections $t_x$ and $t_y$ as $V_x$ and $V_y$, respectively, and $V_r$ for rotation of 90°, $r$, which are defined as follows:

$$V_x = X^{\otimes n} \otimes I^{\otimes n} = X_{1:n} \tag{4}$$

$$V_y = I^{\otimes n} \otimes X^{\otimes n} = X_{n+1:2n} \tag{5}$$

$$V_r = (X^{\otimes n} \otimes I^{\otimes n}) \otimes_{i=0}^{n-1} SWAP_{i:i+n} = V_x V_r^{'} \tag{6}$$

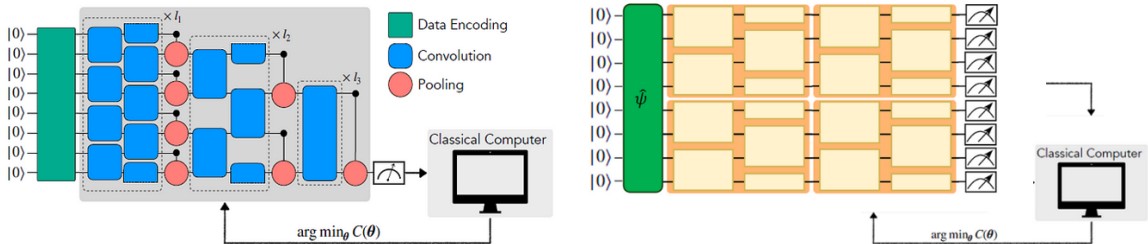

Figure 1: Architecture used to construct left) QCNN and right) EQCNN.

### 2.2.2 Equivariant Ansatz

Once the data is embedded, we apply a quantum circuit designed to be equivariant with respect to roto-reflection symmetries. Using the Twirling Method, we identified a set of quantum gates that preserve these symmetries.

Twirling formula. Let $V_g$ be a unitary representation of G. Then,

$$T_V[X] = \frac{1}{|G|} \sum_{g \in G} V_g X V_g^\dagger \tag{7}$$

defines a projector onto the set of operators commuting with all elements of the representation, i.e.,

$$[T_V[X], V_g] = 0, \text{ for all } X \text{ and } g \in \text{G} \tag{8}$$

This is the same as $\mathcal{U}(\theta)V_g = V_g\mathcal{U}(\theta)$.

Using this formula, we find that the quantum gates that are equivariant are the following:

$$T_V = \{Y1Y2, Z1Z2, X1, X2\}.$$

This is the equivariant gateset that ensures that each gate respects the underlying symmetries of the data, reducing the search space during optimization and improving the efficiency of the model.[1]

Using these quantum gates, we define the $U2\_equiv$ convolutional filter, which serves as the foundation for constructing the equivariant quantum model. To ensure equivariance, we follow the structure outlined in Figure 1, where each yellow block represents a $U2\_equiv$ convolutional filter. It is important to note that the same filter with identical parameters must be applied across all the qubits, a technique known as weight sharing. This convolutional filter has six trainable parameters, and due to weight sharing, each layer utilizes only these six parameters.

### 2.2.3 Invariant Observable

Finally, the quantum state is measured by calculating the expectation values of the Pauli-Z observable for each qubit. [3] These measurements are used for image classification, ensuring that the model's output remains invariant under the symmetries considered. The observable satisfies the condition

$$V_g^\dagger O V_g = O \tag{9}$$

which guarantees its invariance under the group $G$.

## 3   Results

In this work, we trained multiple quantum models utilizing the Mean Square Error (MSE) as the cost function, with the Nesterov optimizer to enhance convergence. [2] The learning rate was set to 0.01, and all models were trained for a total of 200 epochs to ensure sufficient optimization of the parameters with training-test data of 80/20. The experiments were carried out on an Acer Nitro 5

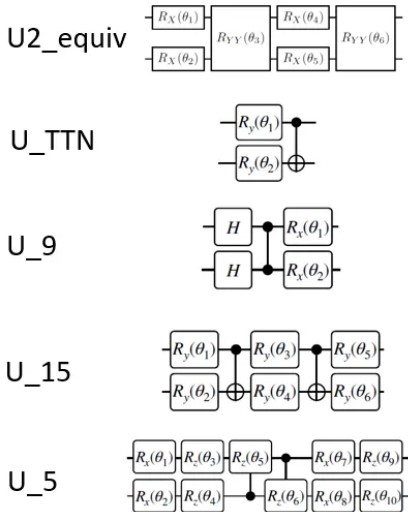

Figure 2: Quantum convolutional filters used to build the equivariant and no-equivariant models.

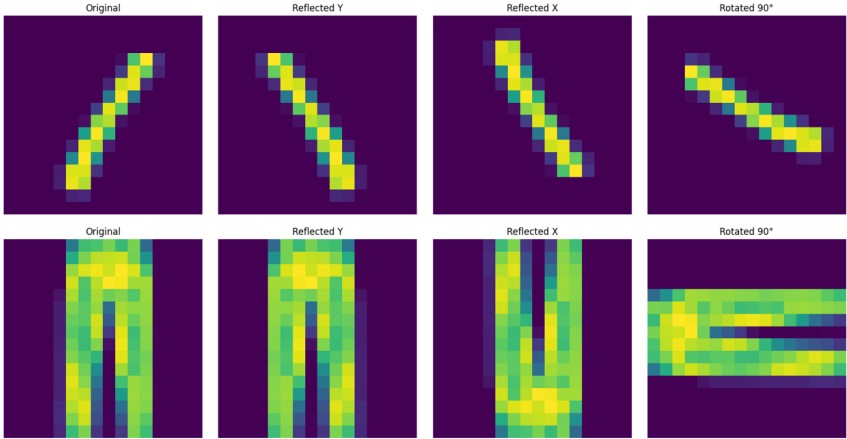

Figure 3: Examples of the symmetry operations that we are considering using up) MNIST and down) Fashion-MNIST datasets.

(2020) laptop, equipped with an Intel Core i5 10th generation processor, 12 GB of RAM, and an Nvidia GeForce GTX 1650 Ti graphics card.

For the equivariant quantum model, we implemented a network architecture composed of three quantum convolutional layers, 18 trainable parameters, designed to maintain symmetry properties. For the other quantum models, we experimented with varying numbers of convolutional filters and layer configurations to explore different feature extraction capabilities.

The entire project was developed using the Pennylane framework, which facilitated the integration of quantum circuits with machine learning techniques. All simulations were executed using quantum simulators, which allowed us to test the models in ideal quantum environments.

A GitHub repository with open-source code and detailed instructions for reproducing the project will be made available and linked here once the work is accepted.

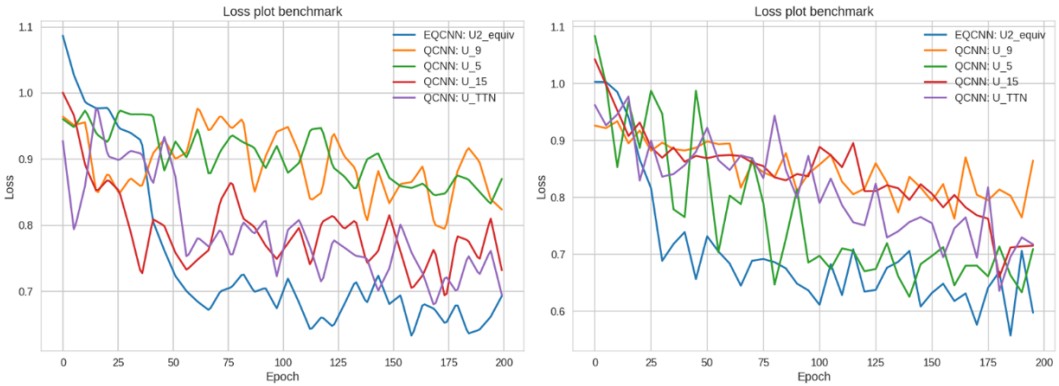

Figure 4: Loss plot comparison among different quantum models. Left) using MNIST. Right) Using Fashion-MNIST.

## 4 Conclusions

By embedding roto-reflection symmetries into our EQCNN, we achieve a more efficient model for image classification and it can take advantage of the NISQ quantum computers era. This approach reduces the parameter space, making the model more data-efficient and improving generalization. We show the effectiveness of our model on benchmark datasets such as MNIST and Fashion-MNIST, demonstrating its potential for applications in quantum machine learning with classical data.

Despite the advances presented, our approach has certain limitations. First, the proposed equivariant quantum model is particularly effective for datasets that exhibit specific symmetries, such as roto-reflections. Its effectiveness may be reduced for datasets that do not display such symmetries. Additionally, as the complexity of the dataset increases or very large datasets are used, scaling the equivariant quantum model becomes more challenging due to the nature of the equivariant ansatz. This limitation may impact the efficiency and performance of the model in broader practical applications.

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
