# OpenReview forum: "Equivariant Quantum Neural Networks for Image Clasification"
_NeurIPS.cc/2024/Workshop/MLNCP — Submitted to MLNCP_

### Official Review · Reviewer_FkCV · 2024-10-04
**Uncompleted working progress work**

**Rating:** 3
**Confidence:** 2

**Review:**

The paper presents an Equivariant Quantum Neural Network for Image classification. The architecture extends QNN, incorporating geometrical symmetries in input data. The paper briefly describes the architecture and some experimental results on a small classification dataset (NMIST and fashion).

# Strengths
- Quantum technologies is a promising technology

# Weaknesses
- Contribution is vague
- Complete missing related works
- Presentation is very poor

# Comments
The paper's contribution is not clear. Equivalent QNN seems to have already been presented in other works like [1]. Moreover, claims like "model is designed to be invariant to certain symmetries in the data; it reduces the number of possible outputs, allowing the model to learn more efficiently" or "weight-sharing, which further reduces the number of parameters that need to be optimized, leading" are not sustained by any evaluation.
- Related work and positioning of the paper to the SOTA is not discussed at all
- Most figures (2,3,4) are not referenced or commented.

# Evaluation
The paper is not complete

[1] West, Maxwell T., Martin Sevior, and Muhammad Usman. "Reflection equivariant quantum neural networks for enhanced image classification." Machine Learning: Science and Technology 4.3 (2023): 035027.

---

### Official Review · Reviewer_hzbM · 2024-10-07
**The method of the authors faces several (well-studied) issues with respect to both classical simulability and scaling that are not addressed in the paper**

**Rating:** 2
**Confidence:** 5

**Review:**

The authors provide an implementation of an equivariant quantum circuit based on "Approximately Equivariant Quantum Neural Network for p4m Group Symmetries in Images" by Chang et al. In particular, their method of performing the circuit construction appears to be a specific implementation of the general method outlined in that work. There are a few issues with the current paper, but the largest is a lack of motivation.

To begin with, there is little evidence in the paper for a scaling advantage over a classical computing approach, and plenty of evidence against in existing literature. From the onset, the suggested encoding appears to use a Hilbert space on the order of the size of the image. Consequently, the cost of classically simulating the protocol is on the order of storing the image digitally. If the authors mean to use a larger Hilbert space via the use of ancillas and more sophisticated encoding schemes, this is not specified in the paper, nor is the total number of qubits explicitly and clearly stated (if I have missed this, I would ask the authors to state this a figure caption, the paragraph with their simulation costs, or somewhere easily detectable -from what they've written, it seems their simulation is on two qubits?).
Even if the authors do plan on using a larger Hilbert space overall, their approach suffers from two other issues, both arising from the fact that the Lie Algebraic dimension of their choice of circuit is polynomially large in the number of qubits. Their choice of observable, single pauli operators on individual qubits, are *outside* the Lie algebra generated by their circuit, implying that for deep circuits and random parameter choice, their training algorithm will experience barren plateaus (see "A Lie Algebraic Theory of Barren Plateaus for Deep Parameterized Quantum Circuits"(2023) by Ragone et al). For shallower circuits, their approach becomes easier to simulate. If the author was to choose an observable set within their Lie Algebra, it becomes possible to classically, efficiently simulate their protocol using GSIM ("Lie-algebraic classical simulations for variational quantum computing" (2023) by Goh et al). In short, much of the protocol seems close to being either clasically simulable or untrainable.
This is unsurprising, as Quantum Convolutional Neural Networks, a mentioned source of inspiration, have been found to be classically efficiently simulable for *most* choices of parameter settings (see "Does provable absence of barren plateaus imply classical simulability? Or, why we need to rethink variational quantum computing" (2024) by Cerezo et al, or , more pointedly, "Quantum Convolutional Neural Networks are (Effectively) Classically Simulable" (2024) by Bermejo et al - though the former paper came out at the start of the year so its omission in discussion is especially puzzling).

The authors *could* object that they choose smart initializations that help them avoid barren plateaus or simulability ("hard" parameters for QCNN simulations), but this is an argument that would need to be demonstrated either theoretically or numerically at a large scale. Certainly,  if they go the numerical route, a similar numerical comparison for state-of-the-art digital models should be included for accuracy (given similar computational resources, for a fair shake). The authors could also try to demonstrate that they expect a minor polynomial scaling advantage or an empirical computational advantage *despite* this classical simulability (see "Interpretable Quantum Advantage in Neural Sequence Learning" (2023) by Anschuetz et al. for inspiration along these lines).

If the authors are not aiming to prove a scaling advantage, perhaps they are trying to prove an advantage in terms of raw compute time (e.g. for extremely large images that are unwieldy to manage through direct digital computation), or energy consumption? In this case the authors would have to present reasonable estimates of these costs on a NISQ device and compare them to the energy/computation time costs for state of the art digital networks to perform with comparable accuracy.

Absent any of these demonstrations (or even arguments) for quantum advantage, it seems that the main (novel) result is that the authors numerically demonstrate an advantage to choosing the inductive bias of the circuit to respect some symmetry of the problem. While it is good to receive numerical confirmation of this, this is already a very well-known fact both in quantum and classical machine learning((see "Theory of overparametrization in quantum neural networks" (2023) by Larocca et al., or "Inductive biases for deep learning of higher-level cognition" (2022) by Goyal and Bengio, or the famous "Geometric Deep Learning" (2021) by Bronstein et al paper, or even some of the authors' own cited works). On its own, I cannot say this is a novel enough of a result to warrant discussion in NeurIPS.

Minor Issues:
The plot demonstrating performance is given in terms of loss. It would be better to present it in terms of classification accuracy, as it would be easier to compare to known accuracy for classical digital networks (a single Linear Layer, for example, can achieve 87%+ accuracy on the full MNIST dataset).

In conclusion, there is a lot of justifiable skepticism for the usefulness of quantum machine learning on classical data in the research community, especially in recent years. I strongly urge the authors to engage with this skepticism directly in the paper when they attempt submission again to this workshop or elsewhere.

---

### Decision · Program_Chairs · 2024-10-10

Reject